# Health literacy among people at risk or with type 2 diabetes in Norwegian primary care—A cross sectional study

Silje Therese Vågenes[1], Marit Graue[1], Jannicke Igland[1,2], Beate-Christin Hope Kolltveit[1,3], Hilde Kristin Refvik Riise [1,4] *

1 Department of Health and Caring Sciences, Western Norway University of Applied Sciences, Bergen, Norway, 2 Department of Global Public Health and Primary Care, University of Bergen, Bergen, Norway, 3 Vossevangen Medical Center, Voss, Norway, 4 Department of Medicine, Haukeland University Hospital, Bergen, Norway

* hkrr@hvl.no

**Data Availability Statement:** Due to personal data protection legislation and legal restrictions related to confidentiality, the data cannot be deposited online as the study participants have not explicitly

## Abstract

### Aims

This study aims to describe health literacy for people at risk of type 2 diabetes and people with type 2 diabetes using the HLS-EU-Q12, and further examine the association between sociodemographic and clinical characteristics, well-being, overall health and quality of life, and health literacy.

### Material and methods

We performed a cross-sectional study among 142 people at risk of type 2 diabetes and 75 people with type 2 diabetes from four primary care clinics in Norway. These data are baseline data from a randomized controlled trial which seeks to evaluate an interprofessional follow-up intervention. Multiple regression analysis was applied to explore associations to health literacy.

### Results

Inadequate health literacy (sum score <33) was found among 30.2% of people at risk of type 2 diabetes and among 25.7% of people with type 2 diabetes. For people at risk of type 2 diabetes, higher level of education was significantly associated with higher health literacy. Better well-being and quality of life was significantly associated with higher health literacy for both groups.

### Conclusions

The primary health care services should pay greater attention to assessing HL, well-being, and quality of life to facilitate the identification of people at risk of type 2 diabetes with insufficient self- management strategies to prevent or delay the development of manifest disease and its complications.

been informed about, nor approved data sharing when the data were gathered in 2019-2021 (see approval from The Regional Committee for Medical and Health Research Ethics South-East Norway (2019/28/REK south-east A, and the processing of personal data from the Norwegian Centre for Research Data (NSD:ID:821994)). Thus, the data in the current study is not publicly available due to ethical regulatory conditions of data usage. However, the ethics committee allows us to include collaborators to participate if the research questions align with the research question and purpose of the study that the participants have been informed about. Thus, persons who wish to be collaborators may contact the last author (hkrr@hvl.no) to put such applications forward to the Regional Committee for Medical and Health Research Ethics South-East Norway for approval (REK). More information on REK may be found at: https://rekportalen.no/#hjem/home.

**Funding:** This study was funded by The Norwegian Nurses Association which has funded a postdoctoral position to develop and conduct the study (ClinicalTrials.gov (ID: NCTT04076384)). The Western Norway University of Applied Sciences has contributed with general fundings. The funders had no role in study design, data collection and analysis, decision to publish, or preparation of the manuscript.

**Competing interests:** The authors have declared that no competing interests exist.

## Introduction

Among people with type 2 diabetes (T2D), individualization of treatment goals and strategies is a key concern in maintaining adequate blood glucose levels and weight, monitoring cardiovascular risk factors, comorbidities and complications [1]. These goals include behavioural change as well as pharmacological treatments to maintain health and quality of life. People at risk of developing T2D or with manifest T2D in Norway are primarily followed up in the primary healthcare services by the general practitioner.

The current guidelines in Norway on overweight and obesity emphasis the important role primary healthcare has in the identification of people with obesity, and thus with risk of T2DM [2]. A retrospective population-based cohort study found that regardless of a genetic risk, being overweight, obese or having a poor lifestyle, are related to T2DM [3]. Globally it is expected that the diabetes prevalence will reach 592–642 million people in the next fifteen years [4, 5]. A Norwegian study showed that the prevalence of T2DM is estimated to be around 216.000 persons, and despite a decreasing incidence of T2DM in Norway, the prevalence continues to rise [6, 7]. Active engagement in self-management of the disease, combined with motivational support and self-confidence, is paramount to healthy lifestyle choices. As the disease poses more serious health risks when untreated people with diabetes need counselling and support from clinicians to self-manage the disease and to ensure an adequate understanding of available health information [8]. This is also important for early identification of those at risk of T2D and to facilitate interventions which can prevent or delay the development of complications associated with T2D [9]. Both International [10] and Norwegian guidelines [11] focus on the importance of education and support to people at risk of and those with T2D.

To self-manage one's health, adequate competencies to make beneficial health-related decisions and actions is fundamental. These abilities, also named "health literacy" (HL), is increasingly recognized and acted upon to give emphasis to the importance of individual's knowledge, skills, and motivation in maintaining good health and well-being [12]. Adequate treatment of the disease and its symptoms is closely related to level of HL [8]. Low HL in people with T2D has been linked to poor diabetes knowledge [8], inadequate glycaemic control and retinopathy [13]. Some studies have also associated low HL with increased risk of T2D [14, 15]. A systematic review from the US found that almost one third of people with T2D had below average HL [16]. Other studies have found that 50–60% of Europeans have low scores on the ability to acquire correct health information about T2D [14]. Few studies have researched HL in participants at risk of T2D. However, some studies have found an association with participants partaking in unhealthy behaviours and low HL [15, 17]. Lastly, in T2D, HL is a key factor in self-care and appropriate understanding of health information, and therefore fundamental for better management of the disease, which can also benefit their quality of life and well-being. This study aims to describe HL for people at risk of T2D and people with T2D using the HLS-EU-Q12, and further examine the association between sociodemographic and clinical characteristics, well-being, overall health and quality of life, and HL.

## Material and methods

This study has a cross-sectional design and analyses baseline data from a randomized controlled trial which seeks to evaluate an interprofessional follow-up intervention among persons at risk of or with T2D, conducted in different primary clinics in four rural areas of Norway (ClinicalTrials.gov ID: NCT04076384). The average population size in these rural areas was about 7000, and the villages where people lives are small on a global perspective.

## Participants

Persons at risk of or with T2D was identified through screening of people who had scheduled appointments at the primary care clinics during an eight-month period in 2019 (02.05.2019–22.12.2019) [18]. In total, 1404 agreed to participate in the screening. The criteria for eligibility for the intervention study were participants at risk of T2D defined by a Finnish Diabetes Risk Score (FINDRISC) cut-off-score ≥15 and/or Body Mass Index (BMI) ≥30, and participants with T2D (Haemoglobin A1c protein (HbA1c) ≥48 mmol/mol/ 6.5%). Criteria for exclusion were participants with type 1 diabetes, severe somatic disease (cancer, heart disease, kidney disease), pregnancy, cognitive impairment, or dementia (Down's syndrome, Alzheimer's) and/ or mental illness (bipolar disease, schizophrenia). Based on these criteria a total of 1137 persons were excluded. After exclusions, 267 patients defined as at risk of developing T2D and 105 patients with diabetes type 2 were left and invited to participate in the intervention. A total of 141 patients (115 people at risk of T2D and 26 people with T2D) declined to participate. Ten people at-risk and four people with T2D withdrew their consent to participate. Thus, a total of 217 patients accepted to participate in the planned intervention, 142 participants at risk of developing T2D and 75 participants with T2D. See also the flow diagram of the study population (Fig 1). Baseline data for these 217 participants were used in the current study.

## Socio-demographics and clinical variables

We collected information on the following socio-demographic variables: age, sex, marital status (married, cohabitant, divorced, single, widow/widower), living situation (living alone,

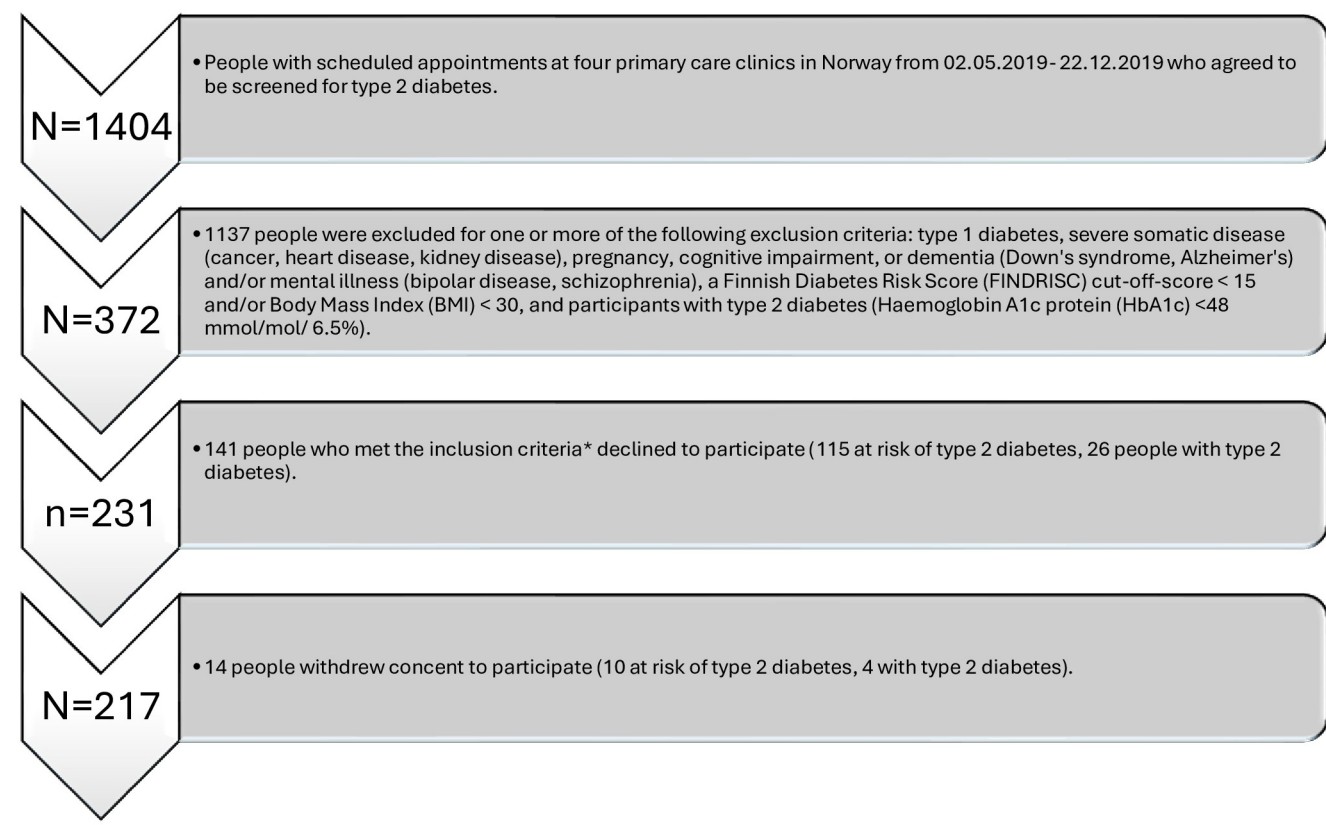

**Fig 1. Flow diagram of the study population.**

living with others), education (primary school/upper secondary school, college/university), employment status (full-time, part-time, retired, other), and activity minimum 30 minutes daily (yes/no). In addition, the following clinical variables were measured: BMI, waist circumference, and HbA1c.

## Measures

The HLS-Q12 was applied to assess HL. The questionnaire is a short version of The European Health Literacy Survey Questionnaire (HLS-EU-Q47) and validated for use in the Norwegian population and for people with T2D [19]. The original HLS-EU-Q47 item scale was by Finbråten and colleges reduced to 12 by combining Rasch modelling and confirmatory factor analysis. For more details of establishment and psychometric properties of the HLS-Q12 see Finbråten et al. 2018 [19]. Shortly, the Norwegian HLS-Q12 is a generic questionnaire with 12 items on a Likert-scale with a 4- point rating scale. The response alternatives vary from very difficult (1), difficult (2), easy (3) and very easy (4), where the higher score sums indicate higher HL proficiencies. The additional response category 'don't know' was recoded to missing. A total score for the individuals was calculated as the mean of the items multiplied by 12 to achieve mean imputation of missing values, giving a score with range 12–48. In addition, cut-offs were applied to divide HL into categories with inadequate (<33), adequate (33–38) and upper normal (>38) HL [20]. It is important to note that the HLS-Q12 used in the current study is different from the HLS19-Q12 [21].

We used the World Health Organization well-being index (WHO-5) to measure subjective psychological well-being [22]. It is a generic five items questionnaire where each item has a score from five (all the time) to zero (none of the time). A total score is calculated as the sum of the items multiplied by four, giving a score with range 25–100. The clinical validity of the questionnaire has been evaluated to be high [22]. It is tested to be psychometrically sound and suitable for diabetes patients [23]. When the WHO-5 is used for the screening of depression, a cut-off score of ≤ 50 is used, whereas a mean score ≤28 is used as a cut-off score for major clinical depression in the primary care setting [24]. In the current study, we divided the WHO-5 sum score into three categories using the 25 and 75 percentiles (lowest tertile (<53), middle tertile (53–77) and highest tertile (≥77).

Two 1-items from The World Health Organization quality of life (WHOQOL) BREF scale [25] was included to measure overall health and quality of life. These two generic questions investigate how the quality of life and health of the participants has been in the previous two weeks. The questions are as following: «How would you assess the quality of your life?" (WHO-Overall QOL), and "How satisfied are you with your health?" (WHO-Overall Health). The response alternatives vary from one (very poor/dissatisfied) to five (very good/satisfied) on a 5-point Likert-scale. The questions have been validated for use in Norway as part of the WHOQOL-BREF [26].

## Statistical analysis

Characteristics of the study population were reported as means and standard deviations for the continuous variables and numbers and percentages for the categorical variables. Differences between persons at risk of T2D and persons with T2D were tested using independent sample t-tests for the continuous variables and a chi-squared test (Person Chi-Square) for the categorical variables.

The participant's continuous HL score was described using mean and standard deviations for subgroups according to various sociodemographic and lifestyle factors, separately for people at risk of T2D and people with T2D. Multiple linear regression analyses were used to

examine the association between the exposure variables (sociodemographic background, clinical variables, well-being, quality of life and general health) and the continuous HL score. Regression models were estimated separately for persons at risk of T2D and persons with T2D. The analyses are shown both unadjusted and adjusted for gender and age using the regression coefficient (B) and 95% confidence interval (CI). The instrument WHO-5 and the items from WHOQOL- BREF were analysed both categorically and as continuous variables. In all analyses the level of significance was defined as a p-value <0.05. Analyses were performed using Stata software (Stata Corp, 2019, Stata Statistical Software: Release 17, College Station, TX: Stata Corp LLC).

## Ethics

This study was approved by the Regional Committee for Medical and Health Research Ethics (REK) in Norway (REK South-East, project number 2019/28), and the Norwegian Agency for Shared Services in Education and Research (SIKT, project number 821994). All participants gave written informed consent.

## Results

### Characteristics of the study population

Table 1 presents sociodemographic and clinical characteristics of the study population. The mean age in the at-risk group was 5.6 years younger than the T2D group (63.5 years, p = 0.006) (numbers not presented in table). Marital status, living situation, and level of education were homogenous between the two groups. Approximately 70% of the participants were married/cohabitant or living with someone. A minor proportion of participants had higher education i.e., a college or university degree (26% in the at-risk group, 19% in the T2D-group). There was a significant difference in HbA1c (p < 0.001) between the two groups. Participants in the at-risk group had a mean HbA1c of 38.0 mmol/mol, while people with T2D had a mean HbA1c of 52.2 mmol/mol. A high proportion of the participants had BMI of ≥30 (59%); 68.3% in the at-risk group and 41.3% in the T2D group.

### Distribution of health literacy scores and associations among sociodemographic background and clinical variables

Distribution of HL scores among the two groups is visualized in Fig 2. A total of 30.2%, 41.3% and 28.6% of the participants in the at-risk group had inadequate (<33), adequate (33–38) and upper normal (>38) HL scores, respectively. The T2D group showed a similar distribution with 25.7% of participants with inadequate HL, 45.7% with adequate HL and 28.6% with upper normal HL.

Associations between sociodemographic background, clinical characteristics and HL can be found in Table 2. No differences in HL were found between the sexes (mean HL sum score of 36.7 and 36.4 in men, compared to 35.5 and 36.2 in women). Moreover, no clear association was found between age and HL in the two groups. The highest HL sum score was found among those 45–54 years of age, while the lowest score was found among those 65 years or older. Further, in the at-risk group, HL increased according to the level of education. Those with a college or university education had a significantly higher score of HL, compared to participants without the same level of education (p = 0.002). There was no significant association between the levels of education and HL in the T2D group. We found no significant association between level of physical activity and HL in either group. In those with T2D, higher BMI (≥ 30) was significantly associated with higher score of HL in the unadjusted analysis

**Table 1. Sample characteristics among people at risk of type 2 diabetes and those with type 2 diabetes.**

| | Type 2 Diabetes (n = 75) | At risk of diabetes (n = 142) | P-value* | Total study population (n = 217) |
|---|---|---|---|---|
| **Gender** | | | 0.073 | |
| Male, n (%) | 38 (50.7) | 54 (38.0) | | 92 (42.4) |
| Female, n (%) | 37 (49.3) | 88 (62.0) | | 125 (57.6) |
| **Age** | | | 0.006 | |
| ≤44 years, n (%) | 4 (5.3) | 22 (15.5) | | 26 (12.0) |
| 45–54 years, n (%) | 9 (12.0) | 36 (25.4) | | 45 (20.7) |
| 55–63 years, n (%) | 22 (29.3) | 31 (21.8) | | 53 (24.4) |
| ≥64 years, n (%) | 40 (53.3) | 53 (37.3) | | 93 (42.9) |
| **Marital status** | | | 0.869 | |
| Unmarried/divorced/widow (er), n (%) | 21 (28.0) | 38 (26.8) | | 59 (27.2) |
| Married/Cohabitant, n (%) | 54 (72.0) | 103 (72.5) | | 157 (72.4) |
| **Living situation** | | | 0.659 | |
| Living alone, n (%) | 18 (24.0) | 31 (21.8) | | 49 (22.6) |
| Living with someone, n (%) | 55 (73.3) | 110 (77.5) | | 165 (76.0) |
| **Education** | | | 0.241 | |
| Primary/Secondary, n (%) | 60 (80.0) | 105 (73.9) | | 165 (76.0) |
| College/university, n (%) | 14 (18.7) | 37 (26.1) | | 51 (23.5) |
| **Employment status** | | | 0.520 | |
| Full-time, n (%) | 23 (30.7) | 48 (33.8) | | 71 (32.7) |
| Part-time n (%) | 10 (13.3) | 20 (14.1) | | 30 (13.8) |
| Retired, n (%) | 34 (45.3) | 49 (34.5) | | 83 (38.3) |
| Other (student, at home, leave of absence), n (%) | 8 (10.7) | 25 (17.6) | | 33 (15.2) |
| **Activity, ≥ 30 min a day** | | | 0.367 | |
| Yes, n (%) | 60 (80.0) | 104 (73.2) | | 164 (75.6) |
| No, n (%) | 10 (13.3) | 25 (17.6) | | 35 (16.1) |
| **Body mass index (BMI)** | | | <0.001 | |
| < 25, n (%) | 10 (13.3) | 8 (5.6) | | 18 (8.3) |
| 25–30, n (%) | 33 (44.0) | 30 (21.1) | | 63 (29.0) |
| ≥ 30, n (%) | 31 (41.3) | 97 (68.3) | | 128 (59.0) |
| **Waist circumference (Male/Female)** | | | 0.037 | |
| ≤ 94/ ≤ 80, n (%) | 5 (6.7) | 1 (0.7) | | 6 (2.8) |
| 94–102 / 80–88, n (%) | 10 (13.3) | 18 (12.7) | | 28 (12.9) |
| ≥ 102/ ≥ 88, n (%) | 57 (76.0) | 117 (82.4) | | 174 (80.2) |
| **HbA1c (mmol/mol), mean (SD)** | 52.2 (9.6) | 38.0 (4.9) | <0.001 | 43.0 (9.7) |

*P-value compares differences among those in risk of type 2 diabetes and those with type 2 diabetes. Chi- square test for categorical variables and t- test for continuous variables. SD indicates, standard deviation; n, number of participants. Percentage of missing: Marital status 0.5%, living situation 1.4%, education 0.5%, activity level 8.3%, BMI 3.7%, waist circumference 4.1%.

(p = 0.047), but not in the adjusted analysis. Neither waist circumference nor HbA1c were significantly associated with HL in the two groups.

## Well-being, quality of life and overall health

Table 3 presents the association between well-being, overall health, quality of life and HL. In the T2D group, high well-being (upper tertile) was positively associated with HL (B = 3.7; p-value 0.019 in the at-risk group and B = 5.9; p-value = 0.002 in the T2D group). The same association was found when well-being was used as a continuous variable (B = 0.1 in both groups,

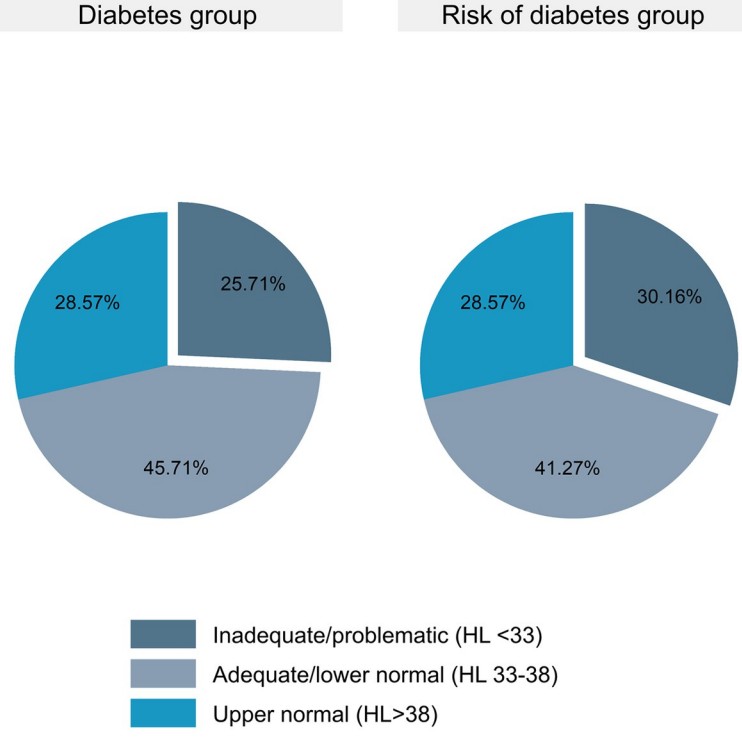

**Fig 2. Health literacy sum score categories stratified by those with type 2 disease and those at risk of type 2 diabetes.**

p-value of 0.003 in the at-risk group, and a p-value of 0.001 in the T2D group). There was a significant association between very good quality of life and HL in the at-risk group (B = 10.8; p = 0.009). There was, however, no significant association between increasing levels of quality of life and HL in the T2D group. When quality of life was used as a continuous variable, we found a significant association with HL in both groups (B = 2.1 in the at-risk group, B = and 1.8 in the T2D-group). When analysing overall health, both categorical and continuous displayed, no significant association with HL was found. At the same time, HL increased with increasing satisfaction with overall health.

## Discussion

To our knowledge, this is the first study to investigate HL in a primary care setting among people at risk of or with T2D. Inadequate health literacy (sum score <33) was found among 30.2% of people at risk of type 2 diabetes and among 25.7% of people with T2D. We found a definite relationship between higher level of education and HL among participants at risk of T2D. Better well-being and quality of life were associated with higher level of HL for both groups.

That higher education was significantly associated with higher HL in the at-risk group is coherent with other findings [27, 28]. This may advocate that interventions for lower educated people with T2D-risk should be prioritised. Lower socioeconomic position such as social status, education and financial resources is considered as a determinant of health that contributes to health inequalities [28]. Current international guidelines emphasise the importance of levelling out societal differences and working for equal health care services for everyone [1]. Diabetes self-management education and support should be emphasised as a fundamental part of the treatment [1].

**Table 2. Associations of health literacy with sociodemographic background and clinical variables, stratified by those at risk of type 2 diabetes and those with type 2 diabetes.**

| | Type 2 Diabetes | | | | | | | At risk of diabetes | | | | | | |
| | Health literacy | | | | | | | Health literacy | | | | | | |
| | n (%) | Mean | SD | B | P-value | B* | P-value* | n (%) | Mean | SD | B | P-value | B* | P-value* |
|---|---|---|---|---|---|---|---|---|---|---|---|---|---|---|
| **Gender** | | | | | | | | | | | | | | |
| Male | 36 (51.4) | 36.4 | 4.8 | Ref. | | | | 46 (36.5) | 36.7 | 4.9 | Ref. | | | |
| Female | 34 (48.6) | 36.2 | 5.6 | -0.1 | 0.918 | -0.2 | 0.889 | 80 (63.5) | 35.5 | 6.0 | -1.2 | 0.247 | -1.2 | 0.232 |
| **Age** | | | | | | | | | | | | | | |
| ≤44 years | 4 (5.7) | 36.5 | 6.2 | Ref. | | | | 20 (15.9) | 35.8 | 5.1 | Ref. | | | |
| 45–54 years | 9 (12.9) | 38.8 | 5.4 | 2.3 | 0.449 | 2.3 | 0.454 | 33 (26.2) | 37.4 | 6.1 | 1.6 | 0.317 | 1.9 | 0.242 |
| 55–63 years | 21 (30.0) | 37.5 | 5.4 | 1.0 | 0.709 | 1.0 | 0.710 | 30 (23.8) | 36.8 | 5.1 | 1.0 | 0.538 | 1.1 | 0.504 |
| ≥64 years | 36 (51.4) | 35.0 | 4.6 | -1.5 | 0.563 | -1.6 | 0.565 | 43 (34.1) | 34.2 | 5.5 | -1.6 | 0.290 | -1.4 | 0.364 |
| **Marital status** | | | | | | | | | | | | | | |
| Unmarried/ divorced/ widow(er) | 20 (28.6) | 36.1 | 5.0 | Ref. | | | | 33 (26.4) | 35.6 | 5.1 | Ref. | | | |
| Married/Cohabitant | 50 (71.4) | 36.4 | 5.3 | 0.3 | 0.830 | 0.2 | 0.907 | 92 (73.6) | 36.1 | 5.8 | 0.5 | 0.649 | 0.1 | 0.925 |
| **Living situation** | | | | | | | | | | | | | | |
| Living alone | 17 (25.0) | 36.0 | 5.7 | Ref. | | | | 27 (21.6) | 36.1 | 5.4 | Ref. | | | |
| Living with someone | 51 (75.0) | 36.6 | 5.1 | 0.6 | 0.698 | 0.4 | 0.774 | 98 (78.4) | 35.9 | 5.7 | -0.3 | 0.816 | -0.7 | 0.564 |
| **Highest level of education** | | | | | | | | | | | | | | |
| Primary/secondary | 55 (79.7) | 36.3 | 5.4 | Ref. | | | | 91 (72.2) | 34.9 | 5.3 | Ref. | | | |
| College/university | 14 (20.3) | 36.6 | 4.3 | 0.3 | 0.834 | 1.0 | 0.546 | 35 (27.8) | 38.7 | 5.4 | 3.8 | 0.000 | 3.5 | 0.002 |
| **Employment status** | | | | | | | | | | | | | | |
| Full-time | 22 (31.4) | 37.8 | 5.7 | Ref. | | | | 44 (34.9) | 37.1 | 5.3 | Ref. | | | |
| Part-time | 10 (14.3) | 35.9 | 4.3 | -1.8 | 0.350 | -1.4 | 0.497 | 19 (15.1) | 34.5 | 6.5 | -2.6 | 0.087 | -2.4 | 0.132 |
| Retired | 30 (42.9) | 35.1 | 4.8 | -2.7 | 0.070 | -0.3 | 0.786 | 41 (32.5) | 34.4 | 5.2 | -2.7 | 0.026 | -0.3 | 0.878 |
| Other^ | 8 (11.4) | 37.3 | 5.5 | -0.5 | 0.806 | -0.1 | 0.911 | 22 (17.5) | 37.7 | 5.5 | 0.6 | 0.664 | 0.9 | 0.557 |
| **Activity level, ≧ 30 min a day** | | | | | | | | | | | | | | |
| Yes | 57 (86.4) | 36.4 | 5.3 | Ref | | | | 93 (80.2) | 36.1 | 5.7 | Ref. | | | |
| No | 9 (13.6) | 35.8 | 4.7 | -0.6 | 0.742 | -1.7 | 0.393 | 23 (19.8) | 36.4 | 5.8 | 0.3 | 0.847 | -0.2 | 0.905 |
| **Body mass index** | | | | | | | | | | | | | | |
| <25 | 10 (14.5) | 34.4 | 5.8 | Ref. | | | | 7 (5.9) | 34.2 | 4.4 | Ref. | | | |
| 25–30 | 29 (42.0) | 35.3 | 4.9 | 0.9 | 0.624 | 1.1 | 0.552 | 24 (20.2) | 36.0 | 4.8 | 1.7 | 0.477 | 0.4 | 0.869 |
| ≥30 | 30 (43.5) | 38.1 | 4.8 | 3.7 | 0.047 | 3.3 | 0.090 | 88 (74.0) | 35.9 | 6.0 | 1.6 | 0.462 | 0.3 | 0.899 |
| **Waist circumference (male/female)** | | | | | | | | | | | | | | |
| <94/ <80 | 5 (7.4) | 34.9 | 4.1 | Ref. | | | | 0 (0.0) | - | - | | | | |
| 94–102 / 80–88 | 10 (14.7) | 35.1 | 5.2 | 0.2 | 0.939 | 0.5 | 0.863 | 17 (14.1) | 36.3 | 4.0 | Ref. | | | |
| >102/ >88 | 53 (77.9) | 36.8 | 5.3 | 1.9 | 0.431 | 2.0 | 0.427 | 104(86.0) | 36.1 | 5.9 | -0.3 | 0.865 | -1.0 | 0.512 |
| **HbA1c** | | | | | | | | | | | | | | |
| <39 mmol/mol (normal) | 3 (4.3) | 35.0 | 2.6 | Ref. | | | | 81 (64.3) | 36.6 | 5.5 | Ref. | | | |
| ≥39–47 mmol/mol (prediabetes) | 23 (32.9) | 36.0 | 4.6 | 1.0 | 0.756 | 0.9 | 0.790 | 39 (31.0) | 34.6 | 5.0 | -2.0 | 0.069 | -1.7 | 0.122 |
| ≥48 mmol/mol (diabetes) | 44 (62.9) | 36.6 | 5.6 | 1.6 | 0.618 | 2.1 | 0.500 | 6 (4.8) | 35.4 | 9.6 | -1.2 | 0.615 | -0.5 | 0.828 |

B = regression coefficient. P-value states significance level between HL and variables listed among those at risk of type 2 diabetes and those with type 2 diabetes.

Calculated with multiple regression analyses.

* Adjusted for gender and age.

^ The category «other» includes students, unemployed, leave of absence.

**Table 3. Associations of health literacy with WHO-5, WHO-Overall Quality of Life and WHO-Overall Health, stratified by those at risk of type 2 diabetes and those with type 2 diabetes.**

| | Type 2 Diabetes | | | | | | | At risk of diabetes | | | | | | |
| --- | --- | --- | --- | --- | --- | --- | --- | --- | --- | --- | --- | --- | --- | --- |
| | Health literacy | | | | | | | Health literacy | | | | | | |
| | n (%) | Mean | SD | B | P-value | B* | P-value* | n (%) | Mean | SD | B | P-value | B* | P-value* |
| **WHO-5, wellbeing** | | | | | | | | | | | | | | |
| Lower tertile (<53) | 10 (14.3) | 34.0 | 2.6 | Ref | | | | 41 (34.2) | 34.2 | 5.2 | Ref. | | | |
| Middle tertile (53–77) | 35 (50.0) | 35.4 | 4.6 | 1.4 | 0.431 | 3.0 | 0.102 | 60 (50.0) | 36.6 | 5.5 | 2.3 | 0.040 | 2.1 | 0.070 |
| Upper tertile (>77) | 25 (35.7) | 38.6 | 5.9 | 4.6 | 0.014 | 5.9 | 0.002 | 19 (15.8) | 37.6 | 6.4 | 3.3 | 0.033 | 3.7 | 0.019 |
| Continuous^ | 70 | 36.3 | 5.2 | 0.1 | 0.006 | 0.1 | 0.001 | 120 | 35.9 | 5.6 | 0.1 | 0.004 | 0.1 | 0.003 |
| **WHO–Overall Quality of Life** | | | | | | | | | | | | | | |
| Very poor | 0 (0.0) | - | - | | | | | 2 (1.7) | 29.4 | 0.8 | Ref. | | | |
| Poor | 2 (2.9) | 33.5 | 4.9 | Ref. | | | | 12 (9.9) | 33.9 | 5.6 | 4.5 | 0.262 | 3.2 | 0.440 |
| Neither good nor poor | 15 (21.4) | 34.2 | 3.6 | 0.7 | 0.858 | 1.7 | 0.660 | 30 (24.8) | 34.7 | 4.4 | 5.3 | 0.168 | 4.0 | 0.300 |
| Good | 36 (51.4) | 36.7 | 5.1 | 3.2 | 0.388 | 4.0 | 0.293 | 64 (52.9) | 36.0 | 5.6 | 6.6 | 0.083 | 5.4 | 0.161 |
| Very good | 17 (24.3) | 37.6 | 6.2 | 4.1 | 0.281 | 5.4 | 0.171 | 13 (10.7) | 41.9 | 5.1 | 12.5 | 0.002 | 10.8 | 0.009 |
| Continuous^ | 70 | 36.3 | 5.2 | 1.6 | 0.043 | 1.8 | 0.025 | 121 | 36.0 | 5.7 | 2.2 | 0.000 | 2.1 | 0.000 |
| **WHO–Overall Health** | | | | | | | | | | | | | | |
| Very dissatisfied | 1 (1.4) | 37 | | Ref. | | | | 1 (0.8) | 28.8 | | Ref. | | | |
| Dissatisfied | 9 (12.9) | 33.8 | 3.9 | -3.2 | 0.558 | -1.0 | 0.863 | 40 (32.3) | 35.6 | 5.4 | 6.8 | 0.233 | 4.2 | 0.456 |
| Neither satisfied nor dissatisfied | 28 (40.0) | 36.7 | 4.2 | -0.3 | 0.956 | 2.3 | 0.669 | 38 (30.7) | 35.9 | 5.7 | 7.1 | 0.213 | 4.9 | 0.385 |
| Satisfied | 30 (42.9) | 36.5 | 6.0 | -0.5 | 0.925 | 2.5 | 0.646 | 44 (35.5) | 36.2 | 5.8 | 7.4 | 0.193 | 5.5 | 0.322 |
| Very satisfied | 2 (2.9) | 39.1 | 11.1 | 2.1 | 0.748 | 4.9 | 0.460 | 1 (0.8) | 45 | | 16.2 | 0.043 | 14.1 | 0.076 |
| Continuous^ | 70 | 36.3 | 5.2 | 0.9 | 0.265 | 1.3 | 0.095 | 124 | 35.9 | 5.6 | 0.6 | 0.279 | 1.0 | 0.108 |

P-value states significance level between health literacy and variables listed among those at risk of type 2 diabetes and those with type 2 diabetes. Calculated with multiple regression analysis.

*Adjusted for gender and age.

^ Exposure variables treated as continuous variables in the regression analyses. Increase in health literacy for every increase in WHO-5 (0–100), WHO overall quality (1–5) of life or WHO overall health (1–5). SD indicates, standard deviation; n, number of participants; B = regression coefficient.

Higher education was not associated with HL in the T2D group. Thus, one might speculate that the differences in education might be levelled out in the treatment and follow-up care offered for people with a known T2D diagnosis in primary care clinics. The nurses might have arranged for more encouraging support to patients with T2D as they have a "diagnosis" and consequently in agreement with national guidelines, they might have got a more structured or organized follow-up. As the current study was conducted in rural areas of Norway the contextual factors might have interfered the associations found. These primary care clinics had at the most 5.000 patients located to the clinic, and in such areas in the countryside the nurses tend to know most of the patients attending follow-up. This might have influenced the associations found for patients with T2D as they were followed-up more regularly by the healthcare professionals than those at risk for T2D. People without a diabetes diagnosis might not been seen as regularly in the clinics because of capacity problems in primary care clinics [27]. The foundation of this hypothesis is also underscored by the fact that people undergoing treatment from specialist healthcare service increase their HL over time [29]. Investigating in the nurse's competence and ability to provide self-management support can have a definitive impact on those receiving education, counselling and treatment [29].

A study reporting on the follow-up routines in primary care clinics in Stockholm has shown that routines for screening, treatment and follow-up for prediabetes and screening for

T2D varies widely [9]. To be identified with a diagnosis might facilitate a greater focus towards health promotion. This matter has also been highlighted by Luo and colleagues [17]. It is problematic that people with prediabetes do not know that they are at risk for future health decline, and hereby not engaged as much in healthy lifestyle activities as they probably should to maintain health and well-being [17]. Thus, it might be advantageous to invest into self-management support interventions that may prevent or delay the development of manifest disease and/or its complications. Younger people with substantial risk of T2D could benefit from a more systematic and timelier screening and follow-up in primary health care clinics to improve their knowledge, skills, and motivation in maintaining good health. However, providing routine follow-up for people at risk of diabetes and ensuring sufficient counselling and support from health care workers are not prioritized [17].

Higher well-being and better quality of life was positively associated with HL in both groups. This finding is consistent with other findings pointing towards HL as an essential factor to obtain higher quality of life [30, 31] and that quality of life can be improved through better HL and self-efficacy [31]. Further, it has been shown that quality of life is strongly related to self-care behaviours [30]. The overall indications of these findings underscore that the level of HL has an important influence on patients, as well as being a general societal predictor of quality of life. Specifically, the findings underscore the degree of access and utilization of healthcare services, the relation between patient and healthcare providers, and the quality of self-care behaviours [32]. On the other hand, Lee and colleagues found that HL only had an indirect effect on quality of life [33]. A similar finding pointed out that low HL and worsening quality of life was independently associated [34]. These findings make it difficult to conclude whether low HL results in worse quality of life, or worse quality of life results in low HL, or both.

Promoting more beneficial self- management strategies is a key task for primary care clinics to help prevent T2D and its complications. It is undoubted that extensive weight reduction can give remission of diabetes [35]. Yet, it is worrying that findings from several qualitative studies have shown that physicians can be unsure of how to talk to patients about obesity, and patients report that they experience that weight issue can be difficult to address in the consultations [36, 37]. Discussions about weight loss might therefore be ethically complex [37]. To enable self-management and shared decision-making it is essential that patients have sufficient HL skills. As one in three patients with T2D in the US display limited functional HL the potential for targeted interventions designed to address HL related barriers is noticeable [16].

Because of the cross-sectional design of this study, causal associations cannot be made. Furthermore, the study was conducted in smaller primary care clinics with a total amount of patients below 5000 in the most populated areas s. Thus, the sample might not be representative for more urban populations of primary care clinics in Norway. In addition, these clinics had consented to participate in an intervention study applying a team-based approach in the follow-up care of the patients and might have had a genuine interest or motivation to provide self-management support. Moreover, the inclusion criteria (FINDRISC score and/or BMI>30) in the at-risk group, might explain the higher proportion of people with high BMI and waist circumference compared the T2D group. Also, the sample size was small, which can lower the generalisability of the findings. The intervention was conducted during the COVID-19 pandemic which might explain why 141 patients (115 people at risk of T2D and 26 people with T2D) declined to participate. Unfortunately, no data on non-responders was available, and we cannot rule out that the non-responders were systematically different from the responders. In general, non-responders report lower socioeconomic status and have poorer health than responders [38]. More research is needed to validate the findings in other clinics. Some strengths of the study are inclusion of two cohorts at distinct stages of the disease, and a high response rate. Finally, we used well-known validated instruments which is another

strength. Clinical measures as well as validation of diagnosis was checked in patient medical records which strengthens the study results.

## Conclusion

This study underlines the importance of identifying those with low HL, specifically younger adults at risk of developing T2D and those with low educational level, to prevent or delay the development of manifest diabetes and its complications. Our results suggest that primary healthcare clinics should be more proactive to identify these people. Additionally, high BMI and low HL among people with T2D might be counteracted with more interventions and better follow-up to improve their knowledge and skills promoting better health outcomes.

## Supporting information

**S1 Checklist. STROBE checklist.**
(PDF)

## Acknowledgments

We would like to express gratitude towards the participants that participated in the study.

## Author Contributions

**Conceptualization:** Marit Graue, Jannicke Igland, Beate-Christin Hope Kolltveit.

**Data curation:** Marit Graue, Jannicke Igland, Beate-Christin Hope Kolltveit.

**Formal analysis:** Silje Therese Vågenes, Jannicke Igland, Hilde Kristin Refvik Riise.

**Funding acquisition:** Marit Graue, Beate-Christin Hope Kolltveit.

**Methodology:** Marit Graue, Jannicke Igland, Hilde Kristin Refvik Riise.

**Project administration:** Beate-Christin Hope Kolltveit.

**Software:** Jannicke Igland, Hilde Kristin Refvik Riise.

**Visualization:** Hilde Kristin Refvik Riise.

**Writing – original draft:** Silje Therese Vågenes.

**Writing – review & editing:** Marit Graue, Jannicke Igland, Beate-Christin Hope Kolltveit, Hilde Kristin Refvik Riise.

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
