## [Decision Letter · Decision Letter 0]

23 Jun 2024

PONE-D-24-04040Health literacy among people at risk or with type 2 diabetes in primary carePLOS ONE

Dear Dr. Riise,

Thank you for submitting your manuscript to PLOS ONE. After careful consideration, we feel that it has merit but does not fully meet PLOS ONE’s publication criteria as it currently stands. Therefore, we invite you to submit a revised version of the manuscript that addresses the points raised during the review process.

We look forward to receiving your revised manuscript.

Kind regards,

Filipe Prazeres, MD, MSc, Ph.D.

Academic Editor

PLOS ONE

Journal Requirements:

"This study was funded by The Norwegian Nurses Association which has funded a postdoctoral position to develop and conduct the study (ClinicalTrials.gov (ID: NCTT04076384)). The Western Norway University of Applied Sciences has contributed with general fundings."

3. Thank you for stating the following in your Competing Interests section: "None declared."

4. In the online submission form, you indicated that "The data underlying the results presented in the study are available on resonable request from Beate-Christin Hope Kolltveit. E-mail: Beate-Christin.Hope.Kolltveit@hvl.no."

5. Please amend the manuscript submission data (via Edit Submission) to include author "Dr. Jannicke Igland"

Reviewers' comments:

Reviewer's Responses to Questions

**Comments to the Author**

1. Is the manuscript technically sound, and do the data support the conclusions?

Reviewer #1: Partly

Reviewer #2: Partly

2. Has the statistical analysis been performed appropriately and rigorously? 

Reviewer #1: Yes

Reviewer #2: No

3. Have the authors made all data underlying the findings in their manuscript fully available?

Reviewer #1: No

Reviewer #2: No

4. Is the manuscript presented in an intelligible fashion and written in standard English?

Reviewer #1: Yes

Reviewer #2: Yes

5. Review Comments to the Author

Reviewer #1: Abstract and main text

This study aims to i) describe health literacy for people at risk of or with type 2

diabetes using the Health Literacy European survey (HLS-EU-Q12), ii) examine the

association between sociodemographic and clinical characteristics and health literacy,

iii) examine the association of well-being, overall health and quality of life with health

literacy.

Usually the objective in a manuscript is not decribed in topics, and the authors could rewrite the aim taking off the numbers.

Introduction

The authors emphasized the issue of Health Literacy in the introduction, however it is important to contextualize type 2 diabetes, which is an important disease, which poses more serious health risks when untreated and therefore it is important to manage this condition. In addition, it is necessary providing epidemiological information on diabetes locally and elsewhere in the world that justifies the relevance of this subject.

Material and Methods

The way the authors describe this study “We conducted a cross-sectional study among participants at risk of or with T2D” The way the authors describe the study method makes me think of a control case, patients and non-sick people, although there is no temporality in the investigation.

Also, these people were invited to participate na intervention study, but the data of the presente study is regards of all participants of the intervention?

Throughout the text, the authors describe the selection of the intervention study, with terms such as randomization. But it is important to make it clear: the reason for refusing to participate, whether the participants were assessed in the first moment, before the intervention, or during, as I do not think that all participants were eligible for the cross-sectional study of this sample, I think it is important to describe a little this larger project, whether there was a sample size calculation or study power for this inference.

The outcome was HL and regression analysis were performed for both groups? How was calculated the outcome?

The authors just mentioned that the sample are from a rural área in the discussion but it will be better this information come in the begining and also better discussion and improve the understanding of this contextual factors that may interfere in the associations found.

The authors recognized the study limitions.

Reviewer #2: This is an interesting topic but I think it has a number of significant methodological issues which are limiting. A fairly major reanalysis would be needed. I mention these issues briefly here:

1. The study should be accompanied and follow closely the relevant checklist (STROBE). In particular a flow diagram is needed.

2. The participation rate in the study is very low. Data on those who declined to participate (demographic characteristics) should be provided if possible or otherwise discussion about what characterized participates vs non participants is needed to help understand who is participating in this study and how they relate to the general population of interest

3. Exclusion criteria are not clearly defined and in some cases debatable. For example what is "kidney disease"? What eGFR or other criterion is used? Would it not be relevant to include patients with these common diabetes related complications in a diabetes study?

4. Title, abstract and much of the discussion focuses broadly on health literacy without mentioning the actual local context of the study. The study should be localized, as what is relevant to this population in Norway may not generalize elsewhere.

5. Deidentified tabular data and data dictionaries in a repository is the standard data sharing mechanism. If this is not being done than a better justification is needed for why data must be requested and are not being furnished.

6. The authors use of the HLSQ12 is not in line with how it has been validated for coding as far as I can tell (https://m-pohl.net/sites/m-pohl.net/files/inline-files/Factsheet%20HLS19-Q12.pdf: excellent, sufficient, problematic, inadequate)

7. Analysis categorizes multiple continuous variables which should be left continuous.

8. In the case of validated categorical cutpoints for psychometric instruments , justification/citation that these are standard cutpoints (e.g the HLSQ12 analysis which seems to depart from the standard analysis)

9. Multiple regression models only adjust for age and gender, which seems not particularly useful. Better would be a single expertly constructed model incorporating all the relevant covariables.

6. PLOS authors have the option to publish the peer review history of their article (what does this mean?). If published, this will include your full peer review and any attached files.

Reviewer #1: **Yes: **Marília Jesus Batista

Reviewer #2: **Yes: **Peter Rohloff

---

## [Author Response · Author response to Decision Letter 0]

24 Aug 2024

To the editorial team,

PLOS ONE

Manuscript ID PONE-D-24-04040:" Health literacy among people at risk or with type 2 diabetes in primary care”. Thank you for reviewing our manuscript for publication in PLOS ONE, and for constructive comments from the academic editor and reviewers. We have now revised the manuscript accordingly. Below are the comments from the editor and reviewers (in bold) followed by our responses (in cursive). All page numbers refers to the uploaded pdf file labeled “Revised manuscript with track changes”.

Comments from the academic editor

We have now thoroughly checked all PLOS ONE`S style requirements and edited the manuscript and file names accordingly. 

"This study was funded by The Norwegian Nurses Association which has funded a postdoctoral position to develop and conduct the study (ClinicalTrials.gov (ID: NCTT04076384)). The Western Norway University of Applied Sciences has contributed with general fundings."

The funders had no role in the study, and we have now included an updated statement of this in the cover letter as requested. The manuscript file is also updated. The new text is as followed: “This study was funded by The Norwegian Nurses Association which has funded a postdoctoral position to develop and conduct the study (ClinicalTrials.gov (ID: NCTT04076384)). The Western Norway University of Applied Sciences has contributed with general fundings. The funders had no role in study design, data collection and analysis, decision to publish, or preparation of the manuscript”

3. Thank you for stating the following in your Competing Interests section: "None declared."

The authors have no competing interests, and the online submission form on competing interests are completed.

4. In the online submission form, you indicated that "The data underlying the results presented in the study are available on reasonable request from Beate-Christin Hope Kolltveit. E-mail: Beate-Christin.Hope.Kolltveit@hvl.no.

All PLOS journals now require all data underlying the findings described in their manuscript to be freely available to other researchers, either 1. In a public repository, 2. Within the manuscript itself, or 3. Uploaded as supplementary information. This policy applies to all data except where public deposition would breach compliance with the protocol approved by your research ethics board. If your data cannot be made publicly available for ethical or legal reasons (e.g., public availability would compromise patient privacy), please explain your reasons on resubmission and your exemption request will be escalated for approval.

We have addressed the following in our revised cover letter: 

“Due to personal data protection legislation and legal restrictions related to confidentiality, the data cannot be deposited online as the study participants have not explicitly been informed about, nor approved data sharing when the data were gathered in 2019-2021 (see approval from The Regional Committee for Medical and Health Research Ethics South-East Norway (2019/28/REK south-east A, and the processing of personal data from the Norwegian Centre for Research Data (NSD:ID:821994)). Thus, the data in the current study is not publicly available due to ethical regulatory conditions of data usage. However, the ethics committee allows us to include collaborators to participate if the research questions align with the research question and purpose of the study that the participants have been informed about. Thus, persons who wish to be collaborators may contact the last author (hkrr@hvl.no) to put such applications forward to the Regional Committee for Medical and Health Research Ethics South-East Norway for approval (REK). More information on REK may be found at: https://rekportalen.no/#hjem/home.”

 5. Please amend the manuscript submission data (via Edit Submission) to include author "Dr. Jannicke Igland"

The manuscript submission data are now amended to include Jannicke Igland as an author.

All captions for our supporting information are now included at the end of our manuscript. The supporting information includes ethical approval in English and Norwegian, and the STROBE checklist. There is no need for these to be cited in the text. 

Review Comments to the Author

Reviewer #1: Abstract and main text

This study aims to i) describe health literacy for people at risk of or with type 2

diabetes using the Health Literacy European survey (HLS-EU-Q12), ii) examine the

association between sociodemographic and clinical characteristics and health literacy, iii) examine the association of well-being, overall health and quality of life with health literacy. Usually the objective in a manuscript is not decribed in topics, and the authors could rewrite the aim taking off the numbers.

We acknowledge your comment and have changed the objectives to the following: “We aimed to describe HL for people at risk of T2D and people with T2D using the HLS-EU-Q12, and further examine the association between sociodemographic and clinical characteristics, well-being, overall health and quality of life and HL.” 

Introduction

The authors emphasized the issue of Health Literacy in the introduction, however it is important to contextualize type 2 diabetes, which is an important disease, which poses more serious health risks when untreated and therefore it is important to manage this condition. In addition, it is necessary providing epidemiological information on diabetes locally and elsewhere in the world that justifies the relevance of this subject.

Thank you for this comment. We have now addressed this more in depth in the introduction (page 3).

Material and Methods

The way the authors describe this study “We conducted a cross-sectional study among participants at risk of or with T2D” The way the authors describe the study method makes me think of a control case, patients and non-sick people, although there is no temporality in the investigation.

Also, these people were invited to participate na intervention study, but the data of the presente study is regards of all participants of the intervention?

Throughout the text, the authors describe the selection of the intervention study, with terms such as randomization. But it is important to make it clear: the reason for refusing to participate, whether the participants were assessed in the first moment, before the intervention, or during, as I do not think that all participants were eligible for the cross-sectional study of this sample, I think it is important to describe a little this larger project, whether there was a sample size calculation or study power for this inference.

We acknowledge that the description of the study design was not precisely described in the manuscript. We have made changes throughout the manuscript to make this clearer, e.g. on page 4-5. The description of the study population has also been rephrased to further explain the reason for refusing to participate, whether the participants were assessed in the first moment or the assessment of eligibility for the intervention, As requested by the second reviewer, we have also included a flow diagram in the method section (page 5).

We did a sample size calculation for the intervention study to make sure that we had enough statistical power to estimate the effect of the intervention on the primary outcome (the Patient Activation Scale- 13 items). In the current study where we only used baseline data, we did not do a sample size calculation.

The outcome was HL and regression analysis were performed for both groups? How was calculated the outcome?

The description of calculation of the outcomes is described in the method section on page 6 under “measures”:

 “HLS-Q12t is a generic questionnaire with 12 items on a Likert-scale with a 4- point rating scale. The response alternatives vary from very difficult (1), difficult (2), easy (3) and very easy (4), where the higher score sums indicate higher HL proficiencies. The additional response category ‘don’t know’ was recoded to missing. A total score for the individuals was calculated as the mean of the items multiplied by 12 to achieve mean imputation of missing values, giving a score with range 12-48. In addition, cut-offs were applied to divide HL into categories with inadequate (<33), adequate (33-38) and upper normal (>38) HL”.

In addition, on page 7, under statistical analyses, the following is stated: . 

“Multiple linear regression analyses were used to examine the association between the exposure variables (sociodemographic background, clinical variables, well-being, quality of life and general health) and the continuous HL score. Regression models were estimated separately for persons at risk of T2D and persons with T2D. “ 

We hope that this clarifies your questions. 

The authors just mentioned that the sample are from a rural área in the discussion but it will be better this information come in the begining and also better discussion and improve the understanding of this contextual factors that may interfere in the associations found.

Thank you for this input. We have now rephrased the first paragraph under “Material and Methods” (page 4) ) to clarify the rural context. In addition, we have expanded the discussion (page 16).

The authors recognized the study limitions.

Thank you. 

Reviewer #2: 

This is an interesting topic but I think it has a number of significant methodological issues which are limiting. A fairly major reanalysis would be needed. I mention these issues briefly here:

Thank you for your valuable comments. We have made several changes to the method- and result section based on your comments. See more details below.

1. The study should be accompanied and follow closely the relevant checklist (STROBE). In particular a flow diagram is needed.

The STROBE checklist is included as an attachment. We have also included a flow diagram in the method section (page 5). 

2. The participation rate in the study is very low. Data on those who declined to participate (demographic characteristics) should be provided if possible or otherwise discussion about what characterized participates vs non participants is needed to help understand who is participating in this study and how they relate to the general population of interest.

It is unfortunate that data on the non-participants are not available. As shown in the description of the study population and further outlined in Figure 1, we only have background data for those eligible for the intervention study Thus, this study report data based on the available baseline data for the intervention. We have further clarified this in the discussion section under limitations (page 18).

3. Exclusion criteria are not clearly defined and in some cases debatable. For example what is "kidney disease"? What eGFR or other criterion is used? Would it not be relevant to include patients with these common diabetes related complications in a diabetes study?

We understand your point of view, however, in Norway, people with serious illness/co-morbidities, such as kidney disease, are referred to specialist health care for follow up of their diabetes. Thus, the general practitioners and nurses in primary care clinics do not have the main responsibility for the follow-up of these patient groups. People with serious illness/co-morbidities were therefore not included in the current study where we recruited people from primary care. Information on kidney disease was gather from the journal. 

In the method section, we have added more details regarding the study population and furthermore the exclusion criteria (page 5). 

4. Title, abstract and much of the discussion focuses broadly on health literacy without mentioning the actual local context of the study. The study should be localized, as what is relevant to this population in Norway may not generalize elsewhere.

Thank you for this input. We have revised the manuscript accordingly (page 4 and 16).

5. Deidentified tabular data and data dictionaries in a repository is the standard data sharing mechanism. If this is not being done than a better justification is needed for why data must be requested and are not being furnished.

Thank you for this comment. Please see our response to the editor on the top of the document (page 3). Shortly, our data cannot be made publicly available due to ethical reasons. 

6. The authors use of the HLSQ12 is not in line with how it has been validated for coding as far as I can tell (https://m-pohl.net/sites/m-pohl.net/files/inline-files/Factsheet%20HLS19-Q12.pdf: excellent, sufficient, problematic, inadequate)

We acknowledge that the description of the measurement used (HLS-Q12) is not clear in the manuscript and we have no rephrased (page 5-6). There are different versions of the 12 item scale, and therefore also different calculations of Health Literacy. We have used the one by Finsbråten et al. 2018. This questionnaire is a short version of The European Health Literacy Survey Questionnaire (HLS-EU-Q47), but it is not the same questionnaire as the HLS19-Q12 in the document you are referring to. The version we have used is validated for use in the Norwegian population and for people with T2D. For more details of establishment and psychometric properties of the HLS-Q12 please see Finsbråten et al. 2018: https://pubmed.ncbi.nlm.nih.gov/29954382/. Since we have used the questionnaire developed by Finsbråten et al, we have also used cutoffs suggested by the developers. In reference 14, details of the cut-off used may be found (6d20c34f-137e-49f5-b08c-54682aca0a0b.pdf (researchsquare.com). Our calculations and cut-offs will therefor differ from those reported in https://m-pohl.net/sites/m-pohl.net/files/inline-files/Factsheet%20HLS19-Q12.pdf. 

For more information on the different 12 items scale we also recommend: Measuring Comprehensive, General Health Literacy in the General Adult Population: The Development and Validation of the HLS19-Q12 Instrument in Seventeen Countries - PMC (nih.gov) . In the introduction of this paper the different versions of existing health literacy questionnaires are described.

7. Analysis categorizes multiple continuous variables which should be left continuous.

Use of continuous exposure variables assumes a linear relationship with the outcome (Health literacy), and this is not necessarily fulfilled in our case. E.g., a u-shape association is seen for age and we have therefore chosen to use this as a categorical variable. Furthermore, it is also easier to interpret the clinical relevance when variables such as BMI and HbA1c is categorized with clinical relevant cut-offs. E.g., it is easier to

---

## [Decision Letter · Decision Letter 1]

1 Oct 2024

PONE-D-24-04040R1Health literacy among people at risk or with type 2 diabetes in Norwegian primary care – a cross sectional studyPLOS ONE

Dear Dr. Riise,

Thank you for submitting your manuscript to PLOS ONE. After careful consideration, we feel that it has merit but does not fully meet PLOS ONE’s publication criteria as it currently stands. Therefore, we invite you to submit a revised version of the manuscript that addresses the points raised during the review process.

We look forward to receiving your revised manuscript.

Kind regards,

Filipe Prazeres, MD, MSc, Ph.D.

Academic Editor

PLOS ONE

Journal Requirements:

Reviewers' comments:

Reviewer's Responses to Questions

**Comments to the Author**

1. If the authors have adequately addressed your comments raised in a previous round of review and you feel that this manuscript is now acceptable for publication, you may indicate that here to bypass the “Comments to the Author” section, enter your conflict of interest statement in the “Confidential to Editor” section, and submit your "Accept" recommendation.

Reviewer #1: All comments have been addressed

Reviewer #2: All comments have been addressed

2. Is the manuscript technically sound, and do the data support the conclusions?

Reviewer #1: Yes

Reviewer #2: Yes

3. Has the statistical analysis been performed appropriately and rigorously? 

Reviewer #1: Yes

Reviewer #2: Yes

4. Have the authors made all data underlying the findings in their manuscript fully available?

Reviewer #1: No

Reviewer #2: Yes

5. Is the manuscript presented in an intelligible fashion and written in standard English?

Reviewer #1: Yes

Reviewer #2: Yes

6. Review Comments to the Author

Reviewer #1: The authors have adressed most of comments.

Health literacy among people at risk or with type 2 diabetes in primary care

In line 105 I think the term “randomization” makes a confusion about the study design. I understand that is a cross-sectional from a baseline, but, actually it is not necessary the information about the randomization here, that is not importante to this study.

Why did people decline the participation?

Reviewer #2: (No Response)

7. PLOS authors have the option to publish the peer review history of their article (what does this mean?). If published, this will include your full peer review and any attached files.

Reviewer #1: **Yes: **Marília Jesus Batista

Reviewer #2: **Yes: **Peter Rohloff

---

## [Author Response · Author response to Decision Letter 1]

4 Oct 2024

To the editorial team,

PLOS ONE

Manuscript ID PONE-D-24-04040:" Health literacy among people at risk or with type 2 diabetes in primary care”. Thank you for reviewing our manuscript for publication in PLOS ONE, and for constructive comments. We have now revised the manuscript accordingly. Below are the comments from reviewer 1 (in bold) followed by our responses (in cursive). 

Reviewer 1

The authors have adressed most of comments.

In line 105 I think the term “randomization” makes a confusion about the study design. I understand that is a cross-sectional from a baseline, but, actually it is not necessary the information about the randomization here, that is not importante to this study.

Why did people decline the participation?

Thank you for this input. We agree and have removed the word “randomization” on page 105. Unfortunately we have little information on the reason for why people declined participation. We do know that the four people with manifest diabetes who withdrew their consent already were followed by the specialist health care due to a previous hospital admission.

---

## [Editor Report · Decision Letter 2]

8 Oct 2024

Health literacy among people at risk or with type 2 diabetes in Norwegian primary care – a cross sectional study

PONE-D-24-04040R2

Dear Dr. Riise,

We’re pleased to inform you that your manuscript has been judged scientifically suitable for publication and will be formally accepted for publication once it meets all outstanding technical requirements.

Kind regards,

Filipe Prazeres, MD, MSc, Ph.D.

Academic Editor

PLOS ONE
---

## [Editor Report · Acceptance letter]

16 Oct 2024

PONE-D-24-04040R2 

PLOS ONE

Dear Dr. Riise, 

I'm pleased to inform you that your manuscript has been deemed suitable for publication in PLOS ONE. Congratulations! Your manuscript is now being handed over to our production team.

Kind regards, 

on behalf of

Prof. Filipe Prazeres 

Academic Editor

PLOS ONE